# One-minute sit-to-stand test: Reference values for the Chilean population

**Matías Otto-Yáñez**[1], **Rodrigo Torres-Castro**[2]*, **Marisol Barros-Poblete**[3], **Marcela Barros**[4], **Carola Valencia**[5], **Alex Campos**[6], **Leticia Jadue**[7], **Homero Puppo**[2], **Pamela Serón**[8], **Jordi Vilaró**[9]

**1** Grupo de Investigación en Salud, Funcionalidad y Actividad Física (GISFAF), Kinesiología, Facultad de Ciencias de la Salud, Universidad Autónoma de Chile, Santiago, Chile, **2** Department of Physical Therapy, Faculty of Medicine, University of Chile, Santiago, Chile, **3** Programa Doctorado en Ciencias Médicas, Universidad Austral de Chile, Valdivia, Chile, **4** CESFAM Eduardo Frei, Villa Alemana, Chile, **5** Centro de Salud Dr Miguel Concha, Quillota, Chile, **6** CESFAM Alberto Allende Jones, Talagante, Chile, **7** Escuela de Kinesiología, Universidad de Santiago, Santiago, Chile, **8** Departamento de Ciencias de la Rehabilitación, Universidad de La Frontera, Temuco, Chile, **9** Blanquerna School of Health Sciences, Global Research on Wellbeing (GRoW), Ramon Llull University, Barcelona, Spain

* rodritorres@uchile.cl

## Abstract

### Introduction

The one-minute sit to stand test (1min-STST) is a field test used to assess functional capacity. It is easily implementable and of significant clinical utility; however, no reference values are currently available for the Chilean population. The objective of this study was to establish reference values for the 1min-STST in a healthy Chilean population.

### Methods

A multicenter cross-sectional study involving data collection from six locations in Chile was conducted. Healthy adults between 18 and 80 years of age were recruited. The anthropometric variables, levels of physical activity, smoking status, Borg scale ratings, and number of repetitions during the 1min-STST were recorded. Reference values were determined according to sex and age range.

### Results

Four hundred ninety-nine healthy subjects (57.5% women, n = 287; median height, 1.63 (0.14) m; weight, 72.8 (20) kg; average BMI, 27.3 ± 4.1 kg/m²) were included in the study. The median (and the lower limit of normality (LLN) values) for the 1min-STST in men ranged from 18–29 years, with 38 (LLN 27) repetitions and 23 (LLN 15) repetitions for 70–80 years. For women aged 18–29 years, 38 (LLN 28) repetitions were performed, and for women aged 70–80 years, 24 (LLN 17) repetitions were performed.

### Conclusions

This study established reference values for the healthy adult Chilean population.

**Data availability statement:** All relevant data are within the manuscript and its Supporting information files.

**Funding:** The author(s) received no specific funding for this work.;

**Competing interests:** None of the authors declare any conflict of interest for this research

**Abbreviations:** 1min-STST, one-minute sit-to-stand test; ATS/ERS, American Thoracic Society/European Respiratory Society; BMI, Body mass index; IPAQ, International Physical Activity Questionnaire; LLN, Lower limit of normality; ULN, Upper limit of normality.

## Introduction

In a broad sense, functional capacity is the ability to carry out everyday tasks at home or at school or work with enough energy to enjoy leisure-time activities and meet periods of unusual strain or disease [1]. It encompasses various aspects, such as strength, endurance, flexibility, and speed, all of which influence one's physical performance and overall health [2]. This capacity is fundamental for daily tasks and participation in sports or recreational activities [2].

The evaluation of functional capacity is a crucial process for understanding an individual's level of physical fitness [3], and it can be conducted in both laboratory and field settings [4]. Laboratory methods provide more precise and controlled measurements, whereas field tests are more practical and reflect performance in daily situations [5]. Field tests hold significant value, particularly when resources for laboratory evaluations are limited [6]. They offer a reasonable estimation of functional capacity without requiring expensive or personalized equipment [6]. Additionally, they are better suited for broader populations, such as areas with limited access to laboratory facilities, providing an overall view of the physical status of the population [7].

The 6-minute walk test (6MWT) has been widely used as a standard to assess functional capacity in various population groups because of its simplicity and effectiveness [6]. However, in situations where the 6MWT cannot be performed, for example, due to the lack of a corridor with adequate length, a viable alternative is the one-minute sit-to-stand test (1min-STST) [8]. The 1min-STST is particularly valuable in assessing and monitoring patients with respiratory or cardiovascular conditions, providing insights into their functional capacity and the effectiveness of implemented treatments and therapies [9,10].

The literature provides considerable support for the psychometric properties of using the 1min-STST to quantify functional capacity [11,12]. The psychometric properties of the 1min-STST and other field tests are essential to ensure their validity and reliability. These properties refer to the test's ability to accurately and consistently measure an individual's functional capacity [13].

Therefore, the 1min-STST has gained ground in rehabilitation because it is a useful and easily accessible tool for monitoring patients in primary health care and telerehabilitation, facilitating access to and monitoring patients with physical limitations [14,15]. Through these modalities, field tests such as the 1min-STST can be administered more conveniently and accessibly, enabling remote assessment and tracking of patients' progress over time [16]. This benefits both healthcare professionals and patients, facilitating rehabilitation and improving care quality [16,17].

When field tests such as the 1min-STST are used, reference values are essential for interpreting the obtained results in the appropriate context [18]. These reference values provide points of comparison to determine whether an individual's performance falls within normal ranges for their age group and specific conditions [18]. Currently, the most commonly used reference values are from Strassman et al., who compiled normative data that can serve as a reference for interpreting results in different populations and clinical settings [19]. However, these values were determined in a population with a less sedentary lifestyle and obesity than the Latin American population, overestimating the expected performance for the subjects of our population [20].

While existing reference values can serve as a starting point, it is crucial to consider the variability among local populations and contexts. Factors such as age, sex, physical activity level, chronic diseases, medication use, and physiological state (e.g., pregnancy, fasting, or posture) can influence test results [18]. There are ethnic, anthropometric, sociocultural, and dietary factors that can influence the differences in the results of functional capacity tests in

different populations, which would be another reason to justify having our own reference values [21]. Therefore, establishing local reference values and objectives will allow for a more precise and personalized evaluation of functional capacity and patient progress in their specific environment. We aimed to determine 1min-STST reference values from an adult healthy Chilean population between 18 and 80 years of age. Additionally, we compared the reference values with previous data and analyzed them by physical activity level.

## Methods

### Study design and participants

A cross-sectional study was carried out simultaneously in six centers, three of which were universities and the other three were primary care health centers, across different geographical regions of Chile (Villa Alemana, Quilpué, Talagante, Santiago, Talca, Puerto Montt) from June 2019 to June 2023. This study was approved by the Ethics Committee of the Metropolitan Eastern Health Service (14-06-2019). All the subjects provided written consent. This study was performed following the "Strengthening the Reporting of Observational Studies in Epidemiology" (STROBE) Guidelines [22].

Participants were recruited from the general population. The recruitment strategy was uniform across all centers and involved the dissemination of information through posters on the recruiters' social media platforms, physical posters placed both inside and outside the evaluation centers, and email outreach for individuals who voluntarily expressed interest in participating in the research and who met the inclusion criteria, which were as follows: adults between 18 and 80 years of age self-reported as healthy, defined as individuals without any known significant illnesses relevant to the proposed study, who fell within the normal range of body measurements, such as weight, whose mental state allows them to understand and provide valid consent for participation in the study [23], and who stated that they were able to stand up and sit down in a chair. The exclusion criteria were as follows: BMI ≥ 35; chronic or acute respiratory disease in the last 30 days; acute or chronic musculoskeletal injury; and concomitant cardiac, cerebral, or neuromuscular disease that prevents them from performing the tests and/or presents an inability to understand the instructions.

### Measurements

Each participant was assessed at a single visit in a standardized assessment order. First, anthropometric and demographic characteristics were obtained. For smoking status, participants indicated whether they were a "never smoker," "ex- smoker" or "current smoker." The abbreviated version of the International Physical Activity Questionnaire (IPAQ-SF) was used to assess the physical activity level [24]. It was classified as low, moderate, or high.

The 1min-STST test consisted of a simple movement: standing up from a chair and adopting the bipedal position with their knees in maximum extension [25]. A standard 43–46 cm chair with thoracolumbar support was used [26]. The subjects sat upright on the chair against the wall with the knees and hips flexed and the feet flat on the floor at shoulder width. The test's previous instructions stipulate that the subject must complete the maximum feasible number of repetitions within one minute. The number of times fully seated and standing up in the chair during one minute was recorded as the primary variable, and only one 1min-STST measurement was performed [27]. In addition, at both the beginning and the end of the test, the perception of effort was evaluated through the modified Borg scale [28].

To standardize the measurements, training sessions were conducted, during which all the evaluators were instructed to record three videos demonstrating the execution of the

1min-STST using a designated pilot test individual. The purpose was to ascertain adherence to the study protocol. Once the validation of the three recorded evaluations was confirmed, authorization was granted to start the study measurements.

## Statistical analysis

The data were analyzed via the statistical software SPSS version 25.0 (IBM Corporation, Armonk, NY, USA). The Kolmogorov–Smirnov test was used to verify the data distribution. Numerical variables are presented as the means and standard deviations (SDs), and qualitative variables are presented as frequencies and percentages. A correlation analysis was also performed via Pearson's or Spearman's test for the quantitative variables (age, weight, height, BMI, initial Borg, and final Borg) with the 1min-STST results. To establish the reference values, we used the categories previously used [29]. Sex- and age-specific normative percentiles (2.5rd, 25th, 50th, 75th, and 97.5th) were generated. The participants were categorized into the following age groups: 18–29, 30–39, 40–49, 50–59, 60–69, and 70–80 years.

To examine the relationships between individual performance in the 1min-STST test and the ages of the subjects, we constructed dispersion graphs depicting the performance distributions concerning age and sex. To assess the impact of physical activity on 1min-STST performance, we compared the number of repetitions at different levels of physical activity, as measured by the IPAQ-SF (low, medium, and high). This comparison was conducted via *t* tests or Kruskal–Wallis tests.

The results of the reference values are presented separately by sex and age group. The values proposed by Strassmann (p50) were compared with those obtained from our study population. Comparisons were made between the measured values and the 50th percentile (p50) reported in the Strassmann et al. study, according to sex and age range. In addition, a linear regression of these data was performed. To assess the differences between the two sets of results, a comparison was conducted via *Student's t* test. This statistical analysis aimed to determine whether there were significant discrepancies between the two datasets.

To calculate the sample size on the basis of the total population, we used free University of Granada software. The risk of type 1 error is 5%, and the confidence level is 97%. As a result, the final estimate determined a minimum of 471 participants.

To facilitate the calculation of reference values, multiple linear regression was performed for men and women, with performance on the 1min-STST as the dependent variable and age, height, and weight as predictors. The stepwise method was used to generate the predictive model.

## Results

A total of 506 participants were recruited for the study. Seven subjects were excluded for having a BMI >35 kg/m². Therefore, the results of 499 people were analyzed, comprising 287 women (57.5%) and 212 men (42.5%). Among the participants, 72.1% reported having a low level of physical activity, and 25.3% declared themselves smokers. Further information on the population's characteristics can be found in Table 1.

The correlations between the values of the 1min-STST and age revealed a moderate inverse relationship (r = −0.529; p < 0.001), a weak correlation with Borg pre (r = −0.253; p < 0.001) and Borg post (r = 0.286; p < 0.001), a very low correlation with weight (r = −0.120; p = 0.006) and a very weak correlation with BMI (r = −0.109; p = 0.015). There was no correlation with height (r = −0.019; p = 0.7).

The relationship between individual performance in the 1min-STST test and the age of the subjects is presented in Fig 1. These graphs visually represent how 1min-STST performance varies across different age groups and between male and female participants.

**Table 1. Characteristics of the study population. The data are presented as the mean (standard deviation), median (interquartile range), or frequency (percentage), as appropriate. BMI, Body mass index; IPAQ-SF, International Physical Activity Questionnaire-Short Form; 1-min-STST, One-minute sit-to-stand test.**

| Variable | All (n = 499) | Women (n = 287) | Men (n = 212) |
|---|---|---|---|
| N by age group (%) | | | |
| 18–29 | 132 (26.5) | 67 (23.3) | 65 (30.7) |
| 30–39 | 70 (14.0) | 35 (12.2) | 35 (16.5) |
| 40–49 | 89 (17.8) | 61 (21.3) | 28 (13.2) |
| 50–59 | 67 (13.4) | 43 (15) | 24 (11.3) |
| 60–69 | 72 (14.4) | 45 (15.7) | 27 (12.7) |
| 70–80 | 69 (13.8) | 36 (12.5) | 33 (15.6) |
| Height(m), median (IR) | 1.63 (0.14) | 1.58 (0.10) | 1.70 (0.10) |
| Weight (kg), median (IR) | 72.8 (20) | 67.5 (18) | 80 (17.6) |
| BMI (kg/m², mean (SD) | 27.3 ± 4.1 | 27.3 ± 4.2 | 27.2 ± 3.9 |
| Physical activity level (IPAQ-SF) | | | |
| Low (%) | 360 (72.1) | 216 (75.3) | 144 (67.9) |
| Moderate (%) | 91 (18.2) | 51 (17.8) | 40 (18.9) |
| High (%) | 48 (9.6) | 20 (7.0) | 28 (13.2) |
| Tobacco use | | | |
| Smoker (%) | 126 (25.3) | 81 (28.2) | 45 (21.2) |
| Non smoker (%) | 349 (69.9) | 192 (66.9) | 157 (74.1) |
| Former smoker (%) | 24 (4.8) | 14 (4.9) | 10 (4.7) |
| Leg Fatigue pre 1min-STST (Borg scale), median (IR) | 0 (1) | 0 (1) | 0 (1) |
| Leg Fatigue post 1min-STST (Borg scale), median (IR) | 3 (3) | 4 (2) | 3 (3) |

Table 2 shows the results stratified according to percentiles for different age and sex groups, constituting the reference values. In the analysis by level of physical activity according to the IPAQ-SF, the median values of the number of repetitions were 33 for low and moderate levels and 38 for high levels, and the difference between groups was statistically significant (p < 0.001) (Fig 2).

When Strassmann's values were compared with those in this research, the former was greater, with averages of 41.3 and 34.9, respectively (p < 0.001) (S1 Data). In the linear regression of these data, an $R^2$ value of 0.226 (p < 0.001) was obtained.

**The predictive equations for men:**

$$1\text{min-STST Men} = 43.394 - (\text{age} * 0.319) + (\text{height} * 0.029)$$
$$R^2 = 0.35$$

**Predictive equations for women:**

$$1\text{min-STST Women} = 90.583 - (\text{age} * 0.285) - (\text{height} * 0.272)$$
$$R^2 = 0.26$$

In the equation, age must be provided in years, and height must be provided in centimeters.

## Discussion

This study has established reference values to facilitate the comparative analysis of 1min-STST performance, aiming to enhance the interpretation of clinical outcomes. These values were

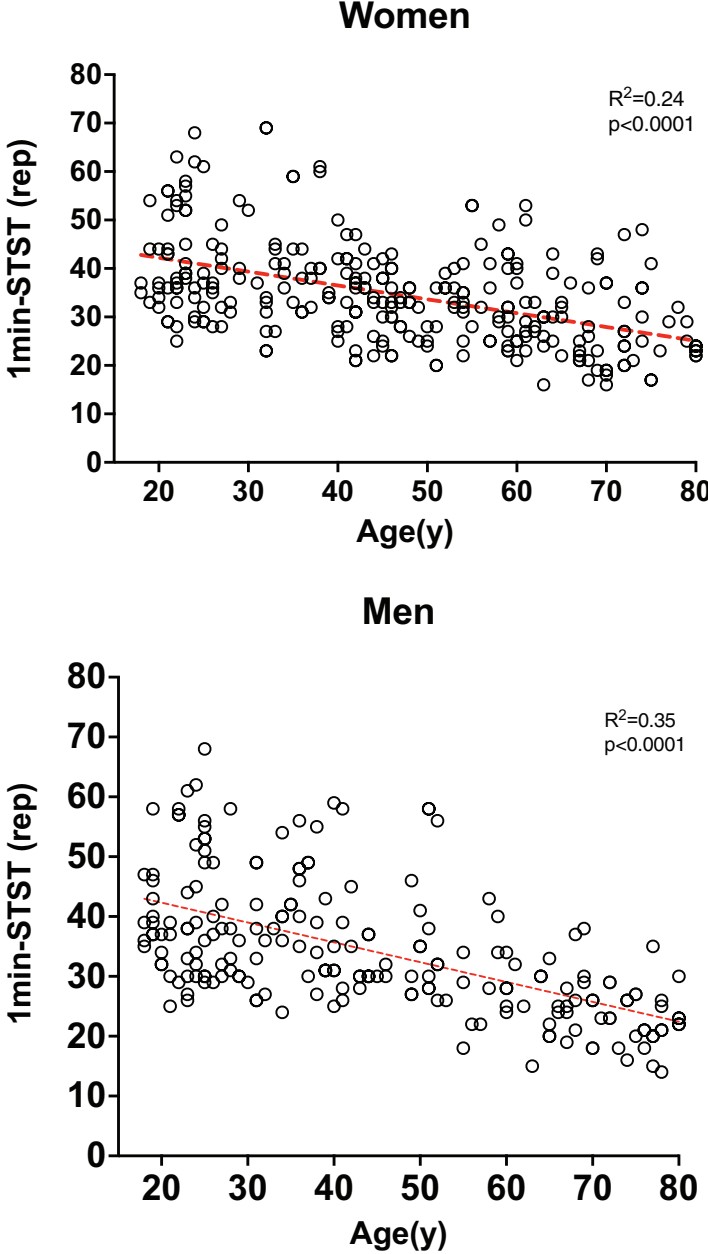

**Fig 1. Association between the number of repetitions of the 1min-STST and age in men and women.** No. of rep, number of repetitions.

derived from a diverse multicentric sample of individuals ranging from 18–80 years old in a developing country with characteristics similar to those of other Latin American populations. The inclusion of such a varied and representative population allows the application of these reference values in similar contexts where local normative data might be lacking. By offering these reference values, this research aims to provide valuable insights and guidance for assessing 1min-STST performance in populations with comparable demographic characteristics, thus assisting healthcare professionals and researchers in their endeavors to evaluate functional capacity in a more contextually relevant manner.

**Table 2. The percentiles are presented according to sex and age categories, along with the delineation of cutoff values and the values denoting normality. Additionally, the lower limit of normal (LLN) at the 2.5th percentile and the upper limit of normal (ULN) at the 97.5th percentile are defined. With the LLN value, we will be able to determine which individuals, according to sex and age, are below the normal range. p50 corresponds to the median, and p25 and p75 correspond to Q1 and Q3, respectively. 1-min-STST: One-minute sit-to-stand test.**

| Age group (years) | Number of 1min-STST repetitions | | | | | | | | | |
|---|---|---|---|---|---|---|---|---|---|---|
| | Women | | | | | Men | | | | |
| | p2.5 | p25 | p50 | p75 | p97.5 | p2.5 | p25 | p50 | P75 | p97.5 |
| 18–29 | 28 | 33 | 38 | 45 | 61 | 27 | 32 | 38 | 47 | 61 |
| 30–39 | 23 | 33 | 38 | 44 | 60 | 26 | 31 | 39 | 47 | 55 |
| 40–49 | 22 | 28 | 33 | 38 | 47 | 26 | 30 | 30 | 37 | 58 |
| 50–59 | 20 | 26 | 32 | 37 | 53 | 20 | 28 | 32 | 39 | 58 |
| 60–69 | 17 | 23 | 28 | 33 | 49 | 15 | 23 | 25 | 30 | 37 |
| 70–80 | 17 | 21 | 24 | 30 | 47 | 15 | 20 | 23 | 26 | 31 |

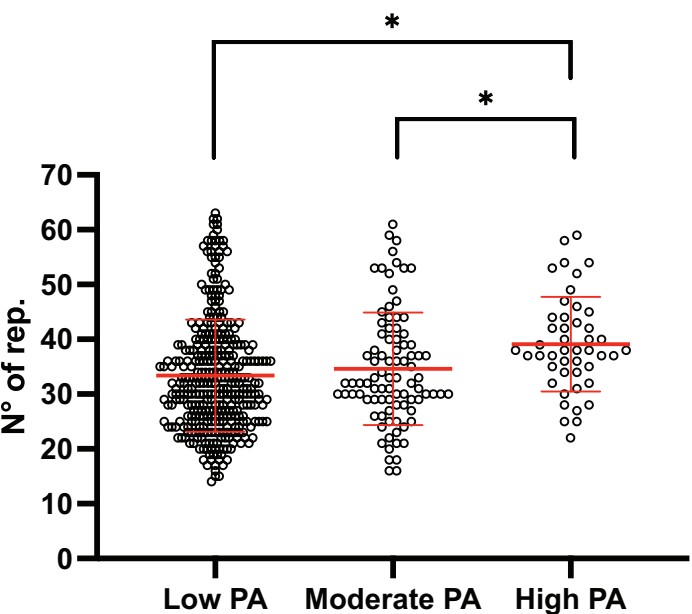

**Fig 2. Comparison of the 1min-STST by physical activity level.**

The importance of reference values adapting to the local population cannot be overstated [18]. When various health parameters and functional capacity are assessed, the use of standardized values based on data from other populations may not accurately reflect the unique characteristics and diversities of the Latin American population.

Our results concerning the influence of physical activity on the number of repetitions are consistent with those of previous reports [30,31]. Our data indicate that individuals with a high level of physical activity perform more repetitions than do those with low and moderate levels of physical activity. This is similar to findings reported by Gürses et al., where healthy subjects with a high level of physical activity according to the IPAQ averaged 50.5 repetitions compared with 41.7 in the moderate physical activity group [30]. This result is also similar to what has been shown in people with diseases [31].

An important aspect of these reference values is the inclusion of subjects with a BMI of 35. The rationale behind this choice stems from the fact that the Latin American population

has significant obesity rates, often surpassing 25% of the total population [20]. Consequently, from a clinical perspective, it is anticipated that between two and three out of every 10 end-users of these tests will present with obesity. Notably, our approach aligns with those of previous studies, as exemplified by Furlanetto et al., who also employed a BMI upper limit of 40 [32].

Furthermore, applying standardized values from other populations might lead to misinterpretations and misdiagnoses, as individual variations may not be adequately accounted for [33]. Therefore, the American Thoracic Society/European Respiratory Society (ATS/ERS) Statement recommends having reference values from the same people on which the procedures will be applied [34].

The results regarding whether a learning effect exists are contradictory. Although some articles suggest its [35] presence while others argue against it [27,36], it is important that if a repetition is performed, strategies should be employed to minimize or nullify this learning effect. Several studies demonstrating no learning effect have actively demonstrated the technical movement to the subject or stopped the test immediately upon detecting errors in execution [37]. Undoubtedly, it is necessary to evaluate whether these strategies, as well as others that may emerge in the future, effectively eliminate this effect.

Another important aspect that lacks consensus is the height of the chair. The literature has reported chair heights between 43 and 48 cm [12,38,39]. In the Chilean context, chairs acquired for healthcare institutions do not have a standard size; however, they tend to range from 43–46 cm. For this reason, to facilitate the execution of the test and the applicability of reference values, especially in primary healthcare settings, we chose this range. Undoubtedly, studies are needed to compare the ideal chair height. However, we believe that more important than the chair height is the distance between the floor and the knee of the subject to establish the ideal height.

An important point to consider is how representative the sample is within the context of the Chilean population. While Strassman et al.'s study included nearly 7,000 subjects [19], reaching such a number is quite challenging, especially in the over 60 age group. Comparatively, in studies from Latin America, Furlanetto et al. evaluated nearly 300 individuals in Brazil [32], a country with a population of 215 million. Our reference values were derived from just under 500 subjects in a population of 18 million. However, the sample size calculation was conducted with a confidence level of 97.5%, and to achieve a 95% confidence level, only 386 subjects were needed. While this may seem like an adequate number, like other reference value studies, it may be representative of adults up to 60 years old, but above that age, it becomes very difficult to find a "healthy" older adult population.

Another point to highlight is that future research should explore what other variables influence the number of repetitions. Our data showed that age is the main factor, as did Strassman's data; however, this explained less than 30% of the test variance [19], suggesting that other parameters could improve the explanation of the 1min-STST variance. Many studies carried out in recent years in our population, especially in post-COVID-19 patients, have used Strassman values [15,40]. Since Swiss values are 30% higher than our values are, there has been a subestimation of functional capacity. This reinforces the idea of having our own values for an adequate characterization of the local population.

Our values are very similar to those shown by Furlanetto et al., who determined the reference values for the 1min-STST in the Brazilian population [32]. For example, the values are the same for women aged 60–69 years (29 repetitions), which is also true for men aged 40–49 years (31 repetitions). However, there is a significant difference from the values presented by Strassman et al. [19] For the same age ranges, Strassman's data show 33 repetitions in women aged 60–69 years, which is 10% higher than our data, and for men aged 40–49 years,

Strassman's expected value is 45 repetitions, a 36% difference. This pattern is consistent and even more pronounced in younger age groups for both sexes, with Strassman's values surpassing our population's values by up to 12 repetitions.

Data previously reported by Strassman revealed higher values in older male adults; however, Furlanetto's values were similar [19,32]. Our values differ from those reported in the literature since the male values are lower. The most likely explanation is possibly selection bias due to the very small sample size in this subgroup.

One limitation of our data is the smaller number of subjects recruited (over 60). This may be because older adults tend to have a sedentary lifestyle and a higher prevalence of non-communicable diseases, making it difficult to obtain a population considered "healthy". The categorization of apparently healthy individuals was determined through self-reports and the absence of diagnosed medical conditions. Nonetheless, it is worth noting that including individuals with undiagnosed conditions or diseases is a prevailing feature in studies involving extensive population samples aimed at establishing reference parameters. Another limitation of this study is that the sampling did not have a probabilistic approach. This could have introduced bias in the selection of the participants since those who accepted the invitation could differ in characteristics from those who did not participate. An aspect not recorded was the socioeconomic characterization that could have influenced the results. Finally, another limitation is that we performed only one repetition. The literature is inconclusive regarding whether this test has a learning effect; however, it appears that in healthy individuals or those without significantly reduced functional capacity, a learning effect may be present, as suggested by Furlanetto et al. [32]. Although only a single measurement was conducted in the present study, all the participants received comprehensive instructions on the correct technique and performed several practice repetitions to ensure proper execution. This approach aimed to mitigate potential bias due to unfamiliarity with the test. The literature presents mixed findings regarding the 1min-STST: some studies, such as that by Vilarinho et al., suggest that multiple repetitions are necessary to accurately assess functional capacity, particularly in younger individuals [41]. In contrast, Bohannon's findings indicate that a single measurement may be sufficient for certain populations, although variability exists depending on the characteristics of the individuals being tested [12]. Therefore, future research should explore strategies to minimize the learning effect, such as implementing standardized practice sessions or conducting additional repetitions, to increase the reliability and validity of test outcomes. Furthermore, there is no consensus on the optimal number of repetitions needed, and it appears that healthy individuals exhibit greater learning effects than do those with chronic conditions.

## Conclusion

Reference values for the 1min-STST were established for a healthy Chilean population aged 18–80 years, providing a simple and inexpensive method for assessing functional capacity. These reference values offer a specific tool for healthcare professionals in Chile and potentially in other countries with similar demographic characteristics to accurately evaluate functional physical capacity. This is particularly important for monitoring patients with chronic diseases, as functional capacity is a key indicator of health status and treatment effectiveness. By using local reference values, healthcare professionals can enhance clinical decision-making, enabling more personalized and effective interventions that ultimately improve patients' quality of life.

## Supporting information

**S1 Data. Database 1minSTST chile.**
(XLSX)

## Author contributions

**Conceptualization:** Matías Otto-Yáñez, Rodrigo Torres-Castro, Marisol Barros-Poblete, Marcela Barros, Carola Valencia, Alex Campos, Leticia Jadue.

**Data curation:** Matías Otto-Yáñez, Marisol Barros-Poblete, Marcela Barros, Carola Valencia, Alex Campos, Leticia Jadue.

**Formal analysis:** Matías Otto-Yáñez, Rodrigo Torres-Castro, Marisol Barros-Poblete, Homero Puppo.

**Investigation:** Matías Otto-Yáñez, Marisol Barros-Poblete.

**Methodology:** Matías Otto-Yáñez, Rodrigo Torres-Castro, Homero Puppo, Pamela Serón, Jordi Vilaró.

**Project administration:** Matías Otto-Yáñez, Marisol Barros-Poblete, Marcela Barros, Carola Valencia, Alex Campos, Leticia Jadue.

**Software:** Matías Otto-Yáñez.

**Supervision:** Rodrigo Torres-Castro, Homero Puppo, Pamela Serón, Jordi Vilaró.

**Visualization:** Matías Otto-Yáñez, Rodrigo Torres-Castro, Marisol Barros-Poblete, Homero Puppo, Pamela Serón, Jordi Vilaró.

**Writing – original draft:** Matías Otto-Yáñez, Rodrigo Torres-Castro.

**Writing – review & editing:** Matías Otto-Yáñez, Rodrigo Torres-Castro, Marisol Barros-Poblete, Marcela Barros, Carola Valencia, Alex Campos, Leticia Jadue, Homero Puppo, Pamela Serón, Jordi Vilaró.

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
