## [Decision Letter · Decision Letter 0]

25 Nov 2024

PONE-D-24-27131One-minute sit-to-stand test: Reference values for Chilean population.PLOS ONE

Dear Dr. Torres-Castro,

Thank you for submitting your manuscript to PLOS ONE. After careful consideration, we feel that it has merit but does not fully meet PLOS ONE’s publication criteria as it currently stands. Therefore, we invite you to submit a revised version of the manuscript that addresses the points raised during the review process.

We look forward to receiving your revised manuscript.

Kind regards,

Luciana Labanca

Academic Editor

PLOS ONE

Journal Requirements:

2. Thank you for stating the following in your Competing Interests section: [None of the authors declare any conflict of interest for this research]. Please complete your Competing Interests on the online submission form to state any Competing Interests. If you have no competing interests, please state "The authors have declared that no competing interests exist.", as detailed online in our guide for authors at http://journals.plos.org/plosone/s/submit-now This information should be included in your cover letter; we will change the online submission form on your behalf.

3. We note that your Data Availability Statement is currently as follows: [All relevant data are within the manuscript and its Supporting Information files.] Please confirm at this time whether or not your submission contains all raw data required to replicate the results of your study. Authors must share the “minimal data set” for their submission. PLOS defines the minimal data set to consist of the data required to replicate all study findings reported in the article, as well as related metadata and methods (https://journals.plos.org/plosone/s/data-availability#loc-minimal-data-set-definition). For example, authors should submit the following data: - The values behind the means, standard deviations and other measures reported; - The values used to build graphs; - The points extracted from images for analysis. Authors do not need to submit their entire data set if only a portion of the data was used in the reported study. If your submission does not contain these data, please either upload them as Supporting Information files or deposit them to a stable, public repository and provide us with the relevant URLs, DOIs, or accession numbers. For a list of recommended repositories, please see https://journals.plos.org/plosone/s/recommended-repositories. If there are ethical or legal restrictions on sharing a de-identified data set, please explain them in detail (e.g., data contain potentially sensitive information, data are owned by a third-party organization, etc.) and who has imposed them (e.g., an ethics committee). Please also provide contact information for a data access committee, ethics committee, or other institutional body to which data requests may be sent. If data are owned by a third party, please indicate how others may request data access.

Reviewers' comments:

Reviewer's Responses to Questions

**Comments to the Author**

1. Is the manuscript technically sound, and do the data support the conclusions?

Reviewer #1: Yes

Reviewer #2: Yes

2. Has the statistical analysis been performed appropriately and rigorously? 

Reviewer #1: Yes

Reviewer #2: Yes

3. Have the authors made all data underlying the findings in their manuscript fully available?

Reviewer #1: Yes

Reviewer #2: Yes

4. Is the manuscript presented in an intelligible fashion and written in standard English?

Reviewer #1: No

Reviewer #2: No

5. Review Comments to the Author

Reviewer #1: Dear authors,

I had the opportunity to review the paper entitled “One-minute sit-to-stand test: Reference values for Chilean population”.

After reading the manuscript I have some concerns that should be resolved:

Abstract:

Authors could specify their conclusion and clinical importance

Introduction:

-It is too long, you can try to combine some sentences like line 77-79

-Please explain the "specific health conditions" in line 126-127

Methods:

- Line 168: You can change it to IPAQ-SF, because you used this term later without abbreviation.

- Line 220: Why did you choose %97 for confidence level?

- Line 179: Please correct “[26]..”

-How did you organize the separation of the participants to the age groups?

Results:

- Use “moderate” istead of moderated

Discussion:

- Consider adding more detail about learning effect

- Which results Socioeconomic characterisation can influence?

Reviewer #2: Some errors were found in the English writing, so I suggest that the article be sent for review before final submission.

The inclusion criteria include adults between 18 and 80 who self-reported as healthy. How do you evaluate healthy considering that 150 of the 499 participants are smokers or former smokers? Do you think that this could have biased your results?

6. PLOS authors have the option to publish the peer review history of their article (what does this mean? ). If published, this will include your full peer review and any attached files.

**Do you want your identity to be public for this peer review?** For information about this choice, including consent withdrawal, please see our Privacy Policy .

Reviewer #1: No

Reviewer #2: No

---

## [Author Response · Author response to Decision Letter 0]

6 Dec 2024

Reviewer 1:

- Authors could specify their conclusion and clinical importance

R: Thank you very much for the suggestion. The conclusion has been modified to provide further detail as requested.

Introduction:

-It is too long, you can try to combine some sentences like line 77-79

R: Thank you very much for the observation. Some parts of the introduction were merged to reduce its length.

-Please explain the "specific health conditions" in line 126-127

R: Thank you for pointing this out. We have revised the sentence to specify the factors directly, such as chronic diseases, medication use, and particular physiological states (e.g., pregnancy, fasting, or posture), which can significantly influence test results. As highlighted by Grasbeck (reference 18), reference values should take into account individual-specific factors to ensure comparability between observed values and reference values.

Methods:

- Line 168: You can change it to IPAQ-SF, because you used this term later without abbreviation.

R: Thank you very much for the correction. The acronym has been changed to IPAQ-SF.

- Line 220: Why did you choose %97 for confidence level?

R: We chose a confidence level of 97% to increase the precision of our population parameter estimates. This decision aimed to reduce uncertainty and obtain a narrower confidence interval. Given the nature of our study, minimizing the risk of misinterpretation of results was particularly important, which justified using a confidence level slightly above the conventional 95%.

- Line 179: Please correct “[26]..”

R: Thank you. The error has been corrected.

-How did you organize the separation of the participants to the age groups?

R: Thank you for the suggestion. The participants were categorized into the following age groups: 18-29, 30-39, 40-49, 50-59, 60-69, and 70-80 years. We have added this information to the methods section

Results:

- Use “moderate” istead of moderated

R: Thank you for the correction. It has been revised to 'moderate'.

Discussion:

- Consider adding more detail about learning effect

R: Thank you very much for the suggestion. The paragraph has been restructured to place greater emphasis on the learning effect.

- Which results Socioeconomic characterisation can influence?

R: Thank you for pointing this out. We acknowledge that the socioeconomic status of participants was not recorded, which could potentially influence the results of the 1min-STST performance. As demonstrated by Shishehbor et al. (2006), lower socioeconomic status is associated with reduced functional capacity, possibly due to limited access to healthcare, nutrition, and opportunities for physical activity. These factors may impact test outcomes, making socioeconomic characterization relevant to better interpret the findings. This limitation has been recognized in the discussion, and we suggest that future studies incorporate socioeconomic characterization to understand its impact on performance outcomes more comprehensively.

Shishehbor, M. H., Litaker, D., Pothier, C. E., & Lauer, M. S. (2006). Association of socioeconomic status with functional capacity, heart rate recovery, and all-cause mortality. Jama, 295(7), 784-792.

Reviewer 2:

- Some errors were found in the English writing, so I suggest that the article be sent for review before final submission.

R: Thank you very much for the suggestion. The manuscript has been sent to a scientific writing editing company for review

- The inclusion criteria include adults between 18 and 80 who self-reported as healthy. How do you evaluate healthy considering that 150 of the 499 participants are smokers or former smokers? Do you think that this could have biased your results?

R: Thank you for raising your concern regarding the inclusion criteria and the classification of participants as 'healthy.' In this study, we applied the Royal College of Physicians' definition of 'healthy,' which considers individuals without significant diseases or functional limitations that could affect their performance in the test to be healthy. Consequently, although some of our participants were smokers or former smokers, they were included as long as they did not present active conditions that could influence their performance. This inclusion also allowed us to better reflect the diversity of the Chilean population, where approximately 30% of adults identify as smokers. Furthermore, we conducted an exploratory comparison between smokers, former smokers, and non-smokers, and found no significant differences in the number of repetitions in the 1min-STST. We have added this information to the Methods section to clarify our approach to classifying participants as healthy.

---

## [Decision Letter · Decision Letter 1]

2 Jan 2025

One-minute sit-to-stand test: Reference values for the Chilean population.

PONE-D-24-27131R1

Dear Dr. Torres-Castro,

We’re pleased to inform you that your manuscript has been judged scientifically suitable for publication and will be formally accepted for publication once it meets all outstanding technical requirements.

Kind regards,

Luciana Labanca

Academic Editor

PLOS ONE

Additional Editor Comments (optional):

Reviewers' comments:

Reviewer's Responses to Questions

**Comments to the Author**

1. If the authors have adequately addressed your comments raised in a previous round of review and you feel that this manuscript is now acceptable for publication, you may indicate that here to bypass the “Comments to the Author” section, enter your conflict of interest statement in the “Confidential to Editor” section, and submit your "Accept" recommendation.

Reviewer #1: All comments have been addressed

2. Is the manuscript technically sound, and do the data support the conclusions?

Reviewer #1: Partly

3. Has the statistical analysis been performed appropriately and rigorously? 

Reviewer #1: Yes

4. Have the authors made all data underlying the findings in their manuscript fully available?

Reviewer #1: Yes

5. Is the manuscript presented in an intelligible fashion and written in standard English?

Reviewer #1: Yes

6. Review Comments to the Author

Reviewer #1: The authors have responded to my comments. All my comments have been adressed. No further questions.

7. PLOS authors have the option to publish the peer review history of their article (what does this mean? ). If published, this will include your full peer review and any attached files.

**Do you want your identity to be public for this peer review?** For information about this choice, including consent withdrawal, please see our Privacy Policy .

Reviewer #1: No

---

## [Editor Report · Acceptance letter]

PONE-D-24-27131R1

PLOS ONE

Dear Dr. Torres-Castro,

I'm pleased to inform you that your manuscript has been deemed suitable for publication in PLOS ONE. Congratulations! Your manuscript is now being handed over to our production team.

Kind regards,

on behalf of

Dr. Luciana Labanca

Academic Editor

PLOS ONE